# OpenReview forum: "MEAL: A Multi-dimensional Evaluation of Alignment Techniques for LLMs"
_ICLR.cc/2026/Conference — Submitted to ICLR 2026_

### Official Review · Reviewer_YAnR · 2025-10-29

**Soundness:** 3
**Presentation:** 1
**Contribution:** 2
**Rating:** 4
**Confidence:** 4

**Summary:**

This paper addresses the critical challenge of evaluating and comparing alignment techniques for Large Language Models (LLMs) to ensure their outputs adhere to human values and safety standards. To tackle this issue, the authors introduce MEAL (Multi-dimensional Evaluation of ALignment Techniques), a unified framework that systematically assesses alignment methods across four key dimensions: alignment detection, alignment quality, computational efficiency, and robustness. Through extensive experiments on various base models and alignment techniques, the paper demonstrates MEAL’s utility by identifying the strengths, limitations, and trade-offs of current state-of-the-art alignment approaches, providing valuable guidance for practitioners in selecting and deploying these techniques.

**Strengths:**

1. The paper addresses a novel and underexplored challenge in the LLM alignment field: how to unify evaluation across multiple alignment dimensions.
2. The work is supported by extensive and well-designed experiments that validate the proposed framework’s effectiveness across diverse settings.
3. The introduction of a unified evaluation framework for LLM alignment is valuable, as it provides essential guidance for researchers and practitioners in selecting and comparing alignment techniques.

**Weaknesses:**

1. The authors lack a detailed description and an overview figure of the key design of the proposed framework in Section 3. The current presentation merely offers a naive explanation of the four evaluation dimensions, which reads more like a group meeting discussion than a rigorous academic framework. A comprehensive framework design should include specifics such as the data design, definitions and extensibility of alignment dimensions, whether new metrics are introduced or existing ones are integrated, and the relationship between the framework’s design and its usage. Additionally, it remains unclear whether the framework supports user customization. An overview figure and a restructured Section 3 are essential to clearly convey the novelty of the work.
2. The relationship between the evaluation protocol and the framework is ambiguous. It is unclear whether the protocols are customizable or fixed, which directly affects the stability and generalizability of the framework. In the current draft, protocols appear to be flexible experimental settings. However, without a clear justification, it is uncertain how changes in protocols might influence the results and, more importantly, the credibility of those results.
3. The experimental section lacks ablation studies on the framework itself. For instance, can additional evaluation dimensions be incorporated based on specific alignment needs? How do different metrics affect the evaluation outcomes? What are the impacts of modifying key design modules within the framework? These questions remain unaddressed, limiting a deeper understanding of the framework’s robustness and adaptability.

**Questions:**

Please check the weakness

---

> ### Author Response · Authors · 2025-12-04
> **Responding to the very valuable comments**
>
> Thank you for your time and effort in reviewing our work. We address the main points below:
>
> **Method organization**
>
> We agree that a detailed description and a visual overview are essential to convey the novelty of our framework more rigorously.
> In the final version, we will restructure Section 3 to provide a clearer and more systematic presentation of the framework’s design. Specifically, we will: expand the description of data design, define and justify the alignment dimensions in detail, clarify the role of metrics, explain the relationship between design and usage and address user customization. The overview figure is already in the appendix.
> Our intention is to demonstrate that the framework is not a naive grouping of dimensions, but a structured, human-centered methodology that integrates models, benchmarks, and human evaluation in a way that existing approaches have not.
>
> **Normalization metrics**
>
> We appreciate your comment regarding normalization. We will include details about the calculation in the manuscript.
>
> ***Normalization and Aggregation of Multi-Dimensional Metrics:*** To enable a unified comparison of alignment techniques across heterogeneous evaluation dimensions, we implemented a normalization procedure followed by an aggregation step to compute a single composite index. Each dimension, produces raw scores derived from distinct methodologies .Therefore, normalization is essential to ensure that all dimensions contribute proportionally to the final index.
>
> ***Normalization procedure:*** For each dimension, we first rescaled the raw metric to the unit interval [0,1][0,1], where 1 represents optimal performance and 0 represents the worst observed performance within the evaluation set. For dimensions where lower raw values indicate better performance, we applied an inversion transformation to align all metrics under a “higher-is-better” paradigm. This guarantees that the composite index consistently reflects improvement as an increase in score across all dimensions.
>
> ***Composite Index:*** After normalization, we computed the overall performance index as the arithmetic mean of the normalized scores across all dimensions.
>
> **Flexibility**
>
> Thank you for raising this important point. We acknowledge that the relationship between the evaluation protocol and the framework needs to be more explicitly articulated. To clarify: the framework is intentionally designed to be modular and customizable, allowing users to adapt protocols based on their specific alignment goals, available resources, and desired evaluation depth.
>
> This flexibility is not arbitrary, it reflects the reality that alignment is multi-faceted and context-dependent. For example, changing the number of evaluation dimensions or the depth of human input will naturally yield different results. This is not a flaw but a feature: it allows researchers to explore trade-offs between bias mitigation, resource constraints, and evaluation richness. We will revise the manuscript to make this clearer and provide justification for how protocol variations influence outcomes, including guidance on how to interpret results under different configurations.
>
> Importantly, while the framework supports customization, it also offers structured defaults to ensure stability and comparability when needed. These defaults can serve as baselines, helping researchers maintain credibility and reproducibility even as they explore alternative settings.
>
> We appreciate your feedback and will incorporate these clarifications to strengthen the transparency and generalization of our framework.
>
> **Ablation studies**
>
> We appreciate the Reviewer highlighting the importance of ablation studies in understanding the robustness and adaptability of our framework. We agree that this is a valuable direction and appreciate the opportunity to clarify.
>
> Our framework is designed to be modular and extensible, allowing researchers to incorporate additional evaluation dimensions based on specific alignment goals. This flexibility is intentional: alignment is context-sensitive, and our framework supports customization to reflect different ethical priorities, cognitive demands, or practical constraints.
>
> We agree that exploring different types of modules would be highly valuable, as it could reduce potential biases and make the evaluation pipeline more robust and fair across all models. While our current work relied on fixed modules due to resource and time constraints, we recognize that this choice may advantage or disadvantage certain methods.
>
> To address this point, we will insert a clarification in the revised manuscript explicitly acknowledging this limitation and highlighting the possibility of incorporating diverse strategies in future iterations of the pipeline. We believe this addition will strengthen the paper by making clear that our intent is to provide a flexible framework that can be extended to ensure fairness and robustness in evaluation across different model families.

---

### Official Review · Reviewer_hfSG · 2025-10-30

**Soundness:** 3
**Presentation:** 2
**Contribution:** 2
**Rating:** 4
**Confidence:** 3

**Summary:**

The paper proposes the MEAL framework, which conducts a multi-dimensional evaluation of various alignment strategies for large language models (LLMs), including alignment detection capability, alignment quality, efficiency, robustness, and safety. Through extensive experiments, the paper validates the performance of different strategies (e.g., instruct, aligner, base, few-shot) across multiple benchmarks, emphasizing that a single metric is insufficient to assess overall alignment effectiveness. It further introduces a multi-dimensional framework for comprehensive comparative evaluation.

**Strengths:**

1. The paper systematically integrates multiple dimensions of alignment evaluation, considering not only alignment quality but also efficiency and safety, making it more comprehensive than existing alignment assessments.

2. The experiments are extensive, covering mainstream benchmark datasets and a wide range of models.

**Weaknesses:**

1. The framework primarily focuses on integrating multi-dimensional metrics and conducting experimental comparisons, lacking innovation in new evaluation methods, theoretical foundations, or alignment assessment techniques. It also does not provide clear, actionable guidance for new algorithms, theories, or alignment approaches. While the comprehensive evaluation has value, it is relatively intuitive and lacks profound new insights or theoretical explanations.

2. Although MEAL offers multi-dimensional evaluation, the framework’s quantifiable metrics and normalization methods are not clearly defined, particularly regarding how to integrate different dimensions and handle trade-offs between them. Its guidance for practical scenarios is limited, and it lacks explicit decision-making procedures.

**Questions:**

Please refer to the relevant points in the Weaknesses section. If the authors can provide clarification and improvements, I would be very happy to raise my score.

---

> ### Author Response · Authors · 2025-12-04
> **Responding to the very valuable comments**
>
> Thank you for your time and effort in reviewing our work.
>
> We address the main points below:
>
> **Novelty**
>
> We appreciate your careful review and understand the concern regarding novelty. However, we would like to emphasize that the strength of our work lies in addressing a critical gap: the absence of a systematic, human-informed, multi-dimensional framework for evaluating alignment. While individual metrics exist, they remain fragmented and difficult to compare. Our contribution integrates these dimensions into a coherent structure, enabling reproducibility, comparability, and transparency across studies.
>
> Rather than proposing yet another isolated evaluation method, our framework provides actionable guidance for researchers: it highlights trade-offs, uncovers gaps in existing approaches, and offers a structured foundation upon which new algorithms and theories can be developed. In this way, our work serves as a bridge between theoretical discussions and empirical practice.
>
> Taken together, these contributions demonstrate that our work is not merely intuitive but offers a structured, human-centered methodology that advances alignment evaluation in ways that existing approaches have not.
>
> **Normalization metrics**
>
> We appreciate your comment regarding normalization. We will include details about the calculation in the manuscript.
>
> ***Normalization and Aggregation of Multi-Dimensional Metrics***: To enable a unified comparison of alignment techniques across heterogeneous evaluation dimensions, we implemented a normalization procedure followed by an aggregation step to compute a single composite index. Each dimension, ***Harm Detection, Alignment Quality, Computational Efficiency***, and ***Robustness and Safety***, produces raw scores derived from distinct methodologies (e.g., classifier accuracy, Likert-scale judgments, latency measurements, and adversarial robustness metrics). These raw scores are inherently non-comparable due to differences in scale, distribution, and interpretability. Therefore, normalization is essential to ensure that all dimensions contribute proportionally to the final index.
>
> ***Normalization procedure:*** For each dimension, we first rescaled the raw metric to the unit interval [0,1][0,1], where 1 represents optimal performance and 0 represents the worst observed performance within the evaluation set. For dimensions where lower raw values indicate better performance, specifically ***Computational Efficiency*** and ***Robustness and Safety***, we applied an inversion transformation to align all metrics under a “higher-is-better” paradigm. This guarantees that the composite index consistently reflects improvement as an increase in score across all dimensions.
>
> ***Composite Index***: After normalization, we computed the overall performance index as the arithmetic mean of the normalized scores across all dimensions. This simple averaging strategy assumes equal importance for all dimensions, providing an interpretable and balanced metric for comparing models and alignment techniques. While alternative weighting schemes or multi-objective optimization approaches could be explored, equal weighting was chosen to avoid introducing subjective bias in the absence of domain-specific prioritization.

---

### Official Review · Reviewer_73Ev · 2025-11-02

**Soundness:** 3
**Presentation:** 3
**Contribution:** 3
**Rating:** 4
**Confidence:** 4

**Summary:**

This paper introduces MEAL, a multi-dimensional evaluation framework for alignment techniques applied to large language models. MEAL formalizes four evaluation axes: alignment detection, alignment quality, computational efficiency, and robustness and safety. The authors implement MEAL across a diverse set of open-source base models, instruct-tuned variants, and lightweight aligner modules, evaluate on multiple benchmarks including BeaverTails, SafeRLHF, XSTEST-response, TruthfulQA, HarmfulQA and Reward-bench 2, and measure detection metrics, pairwise correction win rates under LLM and reward-model judges, end-to-end latency and peak memory, and resistance to adversarial attack using the StrongREJECT suite. The experimental results highlight trade-offs between approaches, with a small specialized aligner called granite-aligner showing strong detection and quality while remaining efficient, ethical-aligner excelling on some robustness measures, and instruct-tuned models often achieving high quality at higher resource cost. The paper argues for choosing alignment strategies by weighing these multiple dimensions rather than a single metric.

**Strengths:**

The paper tackles an important and timely problem, namely the lack of a unified, praxis-oriented evaluation protocol for alignment techniques, and does so in a way that is both broad and practical. The four-dimension decomposition is intuitive and useful because it mirrors real deployment constraints, explicitly bringing together detection, quality, efficiency and robustness into one decision framework. The experimental scope is substantial for a single paper: multiple model families, a diverse set of benchmarks that cover different harm modalities, two separate judging pipelines including reward-model ensembles, and adversarial tests from StrongREJECT. The use of both judge-model panels and reward-model panels to compare aligned versus original responses is a good design choice because it surfaces evaluator variance and highlights reliability issues. The paper is clear in its motivation and structure, presents results in tables that make cross-dimension comparisons possible, and offers thoughtful discussion about how alignment choices produce trade-offs in practice.

**Weaknesses:**

1. The aggregation into a single MEAL score is described but not formalized. The paper should make explicit the normalization and weighting scheme, motivate it, and include ablations showing how overall rankings change under alternative normalizations and weights.
2. The similarity-based detection rule for some aligners is under-specified and potentially brittle. The paper uses BLEU and ROUGE thresholds to decide safety, yet these metrics are known to poorly correlate with semantic equivalence in many cases. Authors can report the exact threshold values, their selection procedure, and sensitivity analyses showing how results change across reasonable thresholds.

**Questions:**

See Weaknesses.

---

> ### Author Response · Authors · 2025-12-03
> **Responding to the very valuable comments**
>
> Thank you for your time and effort in reviewing our work. We address the main points below:
>
> All explanations will be incorporated in the manuscript.
>
> **Normalization metrics**
>
> We appreciate your comment regarding normalization. We will include details about the calculation in the manuscript.
>
> ***Normalization and Aggregation of Multi-Dimensional Metrics***: To enable a unified comparison of alignment techniques across heterogeneous evaluation dimensions, we implemented a normalization procedure followed by an aggregation step to compute a single composite index. Each dimension, ***Harm Detection, Alignment Quality, Computational Efficiency***, and ***Robustness and Safety***, produces raw scores derived from distinct methodologies (e.g., classifier accuracy, Likert-scale judgments, latency measurements, and adversarial robustness metrics). These raw scores are inherently non-comparable due to differences in scale, distribution, and interpretability. Therefore, normalization is essential to ensure that all dimensions contribute proportionally to the final index.
>
> ***Normalization procedure:*** For each dimension, we first rescaled the raw metric to the unit interval [0,1][0,1], where 1 represents optimal performance and 0 represents the worst observed performance within the evaluation set. For dimensions where lower raw values indicate better performance, specifically ***Computational Efficiency*** and ***Robustness and Safety***, we applied an inversion transformation to align all metrics under a “higher-is-better” paradigm. This guarantees that the composite index consistently reflects improvement as an increase in score across all dimensions.
>
> ***Composite Index***: After normalization, we computed the overall performance index as the arithmetic mean of the normalized scores across all dimensions. This simple averaging strategy assumes equal importance for all dimensions, providing an interpretable and balanced metric for comparing models and alignment techniques. While alternative weighting schemes or multi-objective optimization approaches could be explored, equal weighting was chosen to avoid introducing subjective bias in the absence of domain-specific prioritization.
>
> **BLEU/ROUGE metrics**
>
> This comment was very helpful, for aligner models that do not emit safety or confidence labels, to the best of our knowledge there is currently no standard or widely accepted method for deriving "safe vs. harmful'’ classifications. Our use of BLEU/ROUGE thresholds was a practical workaround to include these models for completeness, rather than omit their detection results entirely. We sincerely agree that these metrics were not designed for this purpose. However, we believe that at least including the models was a way to indicate that there may be other ways to evaluate the models together, even if some do not have the same feature. We therefore ask that the reviewer consider our transparency regarding their limitations.
>
> We now report the threshold ranges we tested and summarize the main findings. For both BLEU and ROUGE-2, we evaluated thresholds spanning a wide range, from 0.1 up to 1.0 in increments such as 0.2, 0.3, and so on. This allowed us to check whether our conclusions depend on any particular cutoff. Across all thresholds tested, we observed the same overall pattern: performance metrics shift gradually as the threshold changes, but the relative behavior of the aligner model remains stable. The main findings, including the ordering of datasets, the comparative difficulty of tasks, and the general performance level of the aligner, do not change. In short, our sensitivity analysis shows that the similarity-based proxy is not excessively brittle with respect to threshold selection. Although the exact numerical values vary, the qualitative conclusions remain consistent across all reasonable thresholds. We will include these ablation details in the revised version and are happy to expand them further if the reviewer wishes.
>
> As a future direction, we plan to evaluate detection using LLMs-as-judges, which would allow more nuanced and validated safety assessments across methods.

---

### Official Review · Reviewer_xx4g · 2025-11-10

**Soundness:** 3
**Presentation:** 2
**Contribution:** 2
**Rating:** 4
**Confidence:** 3

**Summary:**

The paper proposes MEAL, a “multi-dimensional” framework to compare LLM alignment techniques across four axes—detection, quality, efficiency, and robustness—then reports results over several datasets and model families.

**Strengths:**

1. The paper proposed an unified evaluation of diverse alignment strategies which is important and timely.
2. Clear structure and released code link.
3. The paper provides the broad metric coverage including detection/quality/efficiency/robustness with reasonably transparent protocols.

**Weaknesses:**

1. The four dimensions largely mirror standard axes used across recent evaluation work (safety/robustness, win-rate judging, latency/memory). The contribution is primarily an aggregation and specific choice of tools, with little methodological innovation (no new metrics, calibration procedures, or principled aggregation). The “overall score” is a simple normalization/averaging that the paper itself questions.
2. For aligners that don’t emit labels, the paper infers “safe vs. harmful” by string similarity thresholds between input and output BLEU/ROUGE with tau=0.5) This is not a validated proxy for safety classification and can spuriously reward minor paraphrases or penalize faithful neutralization. It also prevents reporting AUC/AUROC, making cross-method comparison uneven.
3. The paper optimizes prompts for base/instruct/ICL models but uses fixed training templates for aligners, which likely advantages aligners on some tasks and disadvantages others. There’s no ablation showing sensitivity to prompt choices or threshold calibration, and no standardized prompt budget across methods.
4. The authors acknowledge a “relatively small number of open-source models,” limited families, and non-exhaustive strategies. This undermines the claimed generality of MEAL and makes the overall scoreboard shown in the Table 5 fragile.

**Questions:**

Please refer to the weakness.

---

> ### Author Response · Authors · 2025-12-03
> **Responding to the very valuable comments**
>
> Thank you for your time and effort in reviewing our work.
> All explanations will be incorporated in the manuscript.
>
> **Normalization metrics**
> We appreciate your comment regarding normalization. We will include details about the calculation in the manuscript.
>
> ***Normalization and Aggregation of Multi-Dimensional Metrics:*** To enable a unified comparison of alignment techniques across heterogeneous evaluation dimensions, we implemented a normalization procedure followed by an aggregation step to compute a single composite index. Each dimension, produces raw scores derived from distinct methodologies .Therefore, normalization is essential to ensure that all dimensions contribute proportionally to the final index.
>
> ***Normalization procedure:*** For each dimension, we first rescaled the raw metric to the unit interval [0,1][0,1], where 1 represents optimal performance and 0 represents the worst observed performance within the evaluation set. For dimensions where lower raw values indicate better performance, we applied an inversion transformation to align all metrics under a “higher-is-better” paradigm. This guarantees that the composite index consistently reflects improvement as an increase in score across all dimensions.
>
> ***Composite Index:*** After normalization, we computed the overall performance index as the arithmetic mean of the normalized scores across all dimensions.
>
> **BLEU/ROUGE metrics**
> This comment was very helpful, for aligner models that do not emit safety or confidence labels, to the best of our knowledge there is currently no standard or widely accepted method for deriving "safe vs. harmful'’ classifications. Our use of BLEU/ROUGE thresholds was a practical workaround to include these models for completeness, rather than omit their detection results entirely. We sincerely agree that these metrics were not designed for this purpose. However, we believe that at least including the models was a way to indicate that there may be other ways to evaluate the models together, even if some do not have the same feature. We therefore ask that the reviewer consider our transparency regarding their limitations.
> We now report the threshold ranges we tested and summarize the main findings. For both BLEU and ROUGE-2, we evaluated thresholds spanning a wide range, from 0.1 up to 1.0 in increments such as 0.2, 0.3, and so on. Across all thresholds tested, we observed the same overall pattern: performance metrics shift gradually as the threshold changes, but the relative behavior of the aligner model remains stable. Our sensitivity analysis shows that the similarity-based proxy is not excessively brittle with respect to threshold selection. Although the exact numerical values vary, the qualitative conclusions remain consistent across all reasonable thresholds. We will include these ablation details in the revised version and are happy to expand them further if the reviewer wishes.
> As a future direction, we plan to evaluate detection using LLMs-as-judges, which would allow more nuanced and validated safety assessments across methods.
>
> **Ablation studies to different templates**
> We sincerely appreciate this observation. We agree that exploring different types of templates for aligners would be highly valuable, as it could reduce potential biases and make the evaluation pipeline more robust and fair across all models. While our current work relied on fixed templates due to resource and time constraints, we recognize that this choice may advantage or disadvantage certain methods. To address this point, we will insert a clarification in the revised manuscript explicitly acknowledging this limitation and highlighting the possibility of incorporating diverse template strategies in future iterations of the pipeline.
>
> **Small number of open-source models**
> We agree with the reviewer that the number of open-source models included in our study is not large. However, our selection was designed to cover different types of aligners rather than exhaustively enumerate all available models. In this way, we aimed to ensure that the evaluation pipeline could handle heterogeneous alignment strategies, including those with and without explicit detection mechanisms. The main contribution of this work is not to establish which alignment model is the "best" in the world, but rather, which is the best within a selection made by a user of the framework. This work aims to develop and present a robust workflow for comparison. By demonstrating how MEAL can consistently evaluate diverse families of aligners under a unified framework, we hope to provide a foundation that future work can extend with broader model coverage and more exhaustive strategies.

---

### Meta-Review · Area_Chair_mUSR · 2026-01-08

**Summary:**

While MEAL is a useful engineering-style consolidation of evaluation axes, the submission does not clear the ICLR bar on methodological contribution and evaluation validity: key parts of the framework (detection proxy + composite scoring) remain insufficiently justified, and several “fairness/comparability” issues are acknowledged but not resolved.

Reviewers converged on four decision-driving concerns: (1) limited novelty—MEAL’s four axes (detection/quality/efficiency/robustness) largely mirror standard evaluation dimensions and the paper reads more like an aggregation/scoreboard than a principled new methodology; (2) questionable detection validity for some aligners—the paper infers “safe vs harmful” using BLEU/ROUGE similarity thresholds when models don’t emit labels, which reviewers view as an unvalidated proxy that can reward superficial paraphrases and makes comparisons uneven (e.g., no AUROC); (3) fragile overall scoring—the single “MEAL score” relies on min–max normalization and simple averaging without sufficient justification or ranking-sensitivity ablations under alternative normalizations/weights; and (4) comparability/generalizability gaps—protocol choices (prompt/template differences across method classes) and a relatively small/limited set of open-source model families make the headline scoreboard and “best method” conclusions feel unstable. These concerns collectively led to a borderline-but-below-threshold assessment despite generally positive views on the paper’s practical motivation and breadth of experiments.

**Reviewer Concerns:**

Addressed (partly, mostly via clarification): The rebuttal explains the normalization/aggregation used for the MEAL score (min–max to
0,1, inversion for “lower-is-better,” then equal-weight averaging), addressing requests to formalize the composite score (xx4g/73Ev/hfSG). It also responds to the BLEU/ROUGE threshold criticism by reporting that they swept thresholds over a wide range and claim qualitative stability, partially addressing “under-specified/brittle threshold” concerns (xx4g/73Ev). Finally, it acknowledges presentation issues and promises to restructure Section 3, clarify modularity/customization, and surface an overview figure (YAnR).

Still outstanding (decision-critical): The core issue that BLEU/ROUGE similarity is a non-validated proxy for safety/harm detection remains unresolved; threshold sweeps don’t fix the conceptual mismatch or the unevenness of detection comparisons (xx4g/73Ev). The rebuttal does not provide the requested rank/sensitivity ablations showing how overall MEAL rankings change under alternative normalizations and weights, leaving the headline “overall score/scoreboard” fragile (xx4g/73Ev/hfSG). Concerns about protocol fairness—different prompt optimization budgets for some model classes vs fixed aligner templates—and limited breadth of open-source families are largely acknowledged as limitations rather than controlled for with ablations, so generality and comparability remain in question (xx4g/YAnR/hfSG).

**Reviewer Scores:**

Reviewer xx4g (4 → likely 4): would appreciate added normalization details and threshold sweeps, but the core worry (BLEU/ROUGE proxy validity + fairness of comparison) likely keeps them at borderline; could move to 5 if they prioritize practicality over rigor.

Reviewer 73Ev (4 → likely 4): asked for explicit normalization/weighting and threshold sensitivity; rebuttal directly supplies these clarifications. Still missing ranking sensitivity, but they might raise to 6 given the added specificity.

Reviewer hfSG (4 → likely 6): explicitly said they’d be happy to raise score with clarification; normalization is clarified, and the “framework value as integration” argument may persuade them to 5, possibly 6 if the revised Section 3/figure substantially improves clarity.

Reviewer YAnR (4 → likely 4): presentation/framework-design concerns are only promised to be fixed (“we will restructure”), and framework-ablation questions remain mostly unanswered; likely stays 4, maybe 5 if the final revision genuinely adds a clearer, more formal Section 3 and concrete customization guidance.

---

### Decision · Program_Chairs · 2026-01-26

Reject